# Manufacturing and Characterization of Functionalized Aliphatic Polyester from Poly(lactic acid) with Halloysite Nanotubes

**DOI:** 10.3390/polym11081314

**Published:** 2019-08-06

**Authors:** Sergi Montava-Jorda, Victor Chacon, Diego Lascano, Lourdes Sanchez-Nacher, Nestor Montanes

**Affiliations:** 1Department of Mechanical and Materials Engineering, Universitat Politècnica de València (UPV), Plaza Ferrándiz y Carbonell 1, 03801 Alcoy, Spain; 2Technological Institute of Materials (ITM), Universitat Politècnica de València (UPV), Plaza Ferrándiz y Carbonell 1, 03801 Alcoy, Spain; 3Escuela Politécnica Nacional, 17-01-2759 Quito, Ecuador

**Keywords:** poly(lactic acid), halloysite nanotubes, mechanical characterization, morphology, thermal characterization

## Abstract

This work reports the potential of poly(lactic acid)—PLA composites with different halloysite nanotube (HNTs) loading (3, 6 and 9 wt%) for further uses in advanced applications as HNTs could be used as carriers for active compounds for medicine, packaging and other sectors. This work focuses on the effect of HNTs on mechanical, thermal, thermomechanical and degradation of PLA composites with HNTs. These composites can be manufactured by conventional extrusion-compounding followed by injection molding. The obtained results indicate a slight decrease in tensile and flexural strength as well as in elongation at break, both properties related to material cohesion. On the contrary, the stiffness increases with the HNTs content. The tensile strength and modulus change from 64.6 MPa/2.1 GPa (neat PLA) to 57.7/2.3 GPa MPa for the composite with 9 wt% HNTs. The elongation at break decreases from 6.1% (neat PLA) down to a half for composites with 9 wt% HNTs. Regarding flexural properties, the flexural strength and modulus change from 116.1 MPa and 3.6 GPa respectively for neat PLA to values of 107.6 MPa and 3.9 GPa for the composite with 9 wt% HNTs. HNTs do not affect the glass transition temperature with invariable values of about 64 °C, or the melt peak temperature, while they move the cold crystallization process towards lower values, from 112.4 °C for neat PLA down to 105.4 °C for the composite containing 9 wt% HNTs. The water uptake has been assessed to study the influence of HNTs on the water saturation. HNTs contribute to increased hydrophilicity with a change in the asymptotic water uptake from 0.95% (neat PLA) up to 1.67% (PLA with 9 wt % HNTs) and the effect of HNTs on disintegration in controlled compost soil has been carried out to see the influence of HNTs on this process, which is a slight delay on it. These PLA-HNT composites show good balanced properties and could represent an interesting solution to develop active materials.

## 1. Introduction

In the last decade, the polymer industry has faced important challenges related to new regulations, increasing concern about environment, petroleum depletion and others. Sustainability has consolidated as a leading force in the development of new high environmentally friendly materials [1,2,3]. Petroleum-derived polymers have, in general, a remarkable effect on the overall carbon footprint, so that many researches have focused on the development of new polymers (thermoplastics, thermosetting and elastomers) with a positive effect on their carbon footprint [4,5,6,7]. These environmentally friendly polymers could be classified into three different groups with different environmental connotations [8]. One group is composed of biodegradable petroleum-derived polymers such as most aliphatic polyesters, i.e., poly(ε-caprolactone)—PCL, poly(butylene succinate)—PBS, poly(glycolic acid)—PGA and so on [9,10,11]. Although they are petroleum-derived polymers, they show interest from an environmental standpoint as they can be disintegrated in controlled compost soil conditions. Another interesting and growing group is that of bio-based and non-biodegradable polymers which include polymers such as poly(ethylene)—PE, poly(ethylene terephthalate)—PET, poly(amides)—PAs and so on, which are obtained from renewable resources but are not biodegradable [12,13,14,15,16]. In fact, they show almost identical properties to their corresponding petroleum-derived counterparts. Their interests in that they can contribute to reduce the carbon footprint as they are obtained from plants that are able to fix CO_2_. Finally, there is an increasing interest on a group of polymers that are both bio-based and biodegradable. This group includes polysaccharides and derivatives, e.g., cellulose, chitin, chitosan, starch and derivatives, among others [17,18,19]. Protein-based polymers, e.g., gluten, soybean and collagen, among others are also included in this group [20,21,22,23]. Finally, bacterial polyesters or poly(hydroxyalkanoates)—PHAs, which derive from bacterial fermentation are gaining interest but up today, their synthesis process and purification is still an expensive process but these materials are thought to be a real alternative to a wide variety of polymers due the high number of poly(esters) that can be synthesized by different bacteria. Among all poly(hydroxyalkanoates), it is worthy to note the key role that poly(hydroxybutyrate)—PHB and its copolymers could acquire in the future [24,25,26,27,28].

Above all these polymers, it is worthy to note the current interest of poly(lactic acid)—PLA, which can be obtained by starch fermentation and, subsequently, polymerization. PLA and its blends have gained a privileged position in the polymer industry as it shows balanced properties (mechanical, thermal, barrier, physical and so on) with an everyday most competitive price [29,30,31]. Today it is possible to find PLA in a wide variety of industrial applications that include automotive, construction and building, packaging, houseware and so on [32,33,34,35,36], Behera et al. [37] proposed that by controlling the manufacturing process, the properties of PLA could be tailored e.g., by lowering M_w_ and the glass transition temperature (T_g_). On the other hand, PLA is one of the most widely used material employed for fused deposition modeling (FDM) for 3D printed parts for whatever industry [38,39,40,41]. In addition to its biodegradability, PLA is resorbable and therefore it finds increasing use in the medicine industry (stents, screws, fixation plates, scaffolds for tissue engineering, surgical suture and so on) [42,43,44,45,46]. In these applications, PLA could also act as a drug carrier to provide controlled drug release [47,48]. Micro and nanoencapsulation have given interesting results for different purposes. This can be accomplished by loading the PLA matrix with encapsulated micro- or nanoparticles. In the recent years, nanotubes (NTs) have gained interest in both pharmacy and medicine applications as they can be used as carriers to deliver different active compounds such as antioxidants, antibiotics, antimicrobials and wound healing, among others [49,50,51]. Different nanotubes have been proposed with this aim and different possibilities. Despite carbon nanotubes (CNTs) have attracted most interest, other nanotubes are being studied as potential drug carriers in medicine and pharmacy [52,53,54,55], e.g., titanate and titania nanotubes (TiNTs) [56,57], hydroxyapatite (HApNTs) [58] and zinc oxide (ZnONTs) [59]. One important drawback of nanotubes is their synthesis process, which is, usually, complex and expensive. Nevertheless, aluminosilicates offer interesting properties as potential carriers. Halloysite nanotubes (HNTs) are naturally occurring nanotubes derived from a particular aluminosilicate structure, composed of an external silicate layer and an internal alumina layer with a different specific volume. This particular structure allows growing nanotubes in the form of rolled silica-alumina layers. They are worldwide available at a competitive price and they can be selectively etched to increase their carrying capacity and above all, they are completely biocompatible [60,61]. HNTs have been loaded into different polymer matrices such as poly(lactic acid), *N*,*N*′-ethylenebis(stearamide) (EBS), poly(propylene), poly(amide) 11, poly(ethylene-co-vinyl acetate) (EVA), high impact poly(styrene) (HIPS) [62,63,64] and different loaded drugs such as insulin, thymol and PEG-hirudin [65,66]. In addition to the bioactivity of these composites, they must fulfill some mechanical, thermal, chemical and physical properties to be used in a particular application [67].

This work explores the effect of halloysite nanotubes (HNTs) on mechanical, thermal and thermomechanical properties of a poly(lactic acid) matrix for potential uses in engineering applications. The effect of the loading amount comprised between 3 and 9 wt% is studied.

## 2. Experimental

### 2.1. Materials

A poly(lactic acid) commercial grade Ingeo 6201D in pellet form was supplied by NatureWorks LLC (Minnesota, USA). This grade possesses a density of 1.24 g cm^−3^ and a melt flow index comprised between 15–30 g/10 min measured at a temperature of 210 °C. Despite this commercial grade is intended for melt spinning of fibers, this melt flow index is also suitable for injection molding and this grade has been previously used as a base material for plasticization, blending and manufacturing of wood plastic composites. Halloysite nanotubes (HNTs) with a chemical composition of Al_2_Si_2_O_5_(OH)_4·_2H_2_O and CAS 1332-58-7, were purchased from Sigma Aldrich (Madrid, Spain). HNTs had an average molecular weight of 294.19 g mol^−1^. The average length was comprised between 1 and 3 μm while the external diameter varied from 30 to 70 nm. The typical morphology of these HNTs is shown in Figure 1 where the typical tubular shape can be detected and even the lumen diameter can be seen by transmission electron microscopy technique, it was performed in a Philips microscope model CM10 (Eindhoven, the Netherlands), the acceleration voltage was set at 100 kV. The HNTs samples were spread in acetone then immersed in an ultrasound bath; afterward, a small drop of this solution was poured onto a carbon grid, it was subjected to solvent evaporation at 25 ℃. From the TEM images, the lumen size of HNTs was obtained. At least 50 measurements were performed to obtain the size distribution. As reported in a previous work by Garcia-Garcia et al. [60], the composition of HNTs is mainly SiO_2_ (53.75%) and Al_2_O_3_ (44.57%) and, in a less extent, some additional oxides at less than 1% (P_2_O_5_, Fe_2_O_3_, SO_3_ and CaO, among others). As it can be seen in Figure 1, HNTs were also characterized by field emission electron microscopy (FESEM) operated at an acceleration voltage of 2 kV. Due to the high hydrophilicity of HNTs, they tended to form aggregates (Figure 1c), which shows an aggregate of dimensions 5.1 × 71 μm^2^. The tubular shape could also be observed by FESEM as it is shown in Figure 1d, in which a single HNT can be observed with an external diameter of 105 nm and a length of about 0.5 μm. Nevertheless, as indicated by Garcia-Garcia et al. [60] the size distribution of HNTs is rather heterogeneous.

Both PLA pellets and HNTs were dried at 60 °C for 24 h in an air-circulating oven to remove residual moisture. Different formulations were prepared containing different HNTs loading as summarized in Table 1. Poly(vinyl acetate)—PVAc was obtained from Sigma-Aldrich (Madrid, Spain), it was used as a compatibilizer due to its high hydrophilicity.

### 2.2. Manufacturing of PLA/HNTs Composites

These formulations were placed in a zipper bag and subjected to an initial homogenization process. The mixtures were fed into a twin-screw co-rotating extruder from DUPRA S.L. (Alicante, Spain) with the following temperature profile: 165 °C (hopper), 170 °C, 175 °C and 180 °C (dye). The rotating speed of the screw was set to 40 rpm and the maximum extruder capacity was set at 60%. The obtained compounds, were pelletized for further processing by injection moulding in a Meteor 270/75 from Mateu & Solé (Barcelona, Spain) with the following temperature profile (from the hopper to the injection nozzle): 170 °C, 180 °C, 190 °C and 200 °C. After this, standard samples for characterization were obtained as can be seeing in Figure 2. As it can be seen, the addition of HNTs contribute to a remarkable change in color from white (neat PLA) to dark brown for the composite with 9 wt% HNTs. Therias et al. [68] have reported identical change in color with increasing HNTs loading.

### 2.3. Thermal Characterization

Thermal characterization was carried out by differential scanning calorimetry (DSC) in a DSC 821 from Mettler-Toledo Inc. (Schwerzenbach, Switzerland). The selected thermal cycled was divided into three different stages: A first heating from 30 °C to 200 °C was followed by a cooling down to 0 °C and, finally, a second heating stage from 0 °C up to 350 °C was scheduled. All the stages were run at a rate of 10 °C min^−1^ in nitrogen atmosphere (66 mL min^−1^). Different parameters were obtained from the second heating cycle, namely, the glass transition temperature (*T*_g_), the cold crystallization peak temperature (*T*_cc_) and enthalpy (Δ*H*_cc_), the melt peak temperature (*T*_m_) and enthalpy (Δ*H*_m_) and the degradation temperature (*T*_d_). The degree of crystallinity (χ_c_%) was calculated following Equation (1).
(1)$$\chi_{cPLA\;}\left(\%\right)=\left[\frac{\left|\Delta H_{m}\right|-\;\left|\Delta H_{cc}\right|}{\left|\Delta H_{100\%}\right|\cdot w_{PLA}}\right]\cdot 100$$
where Δ*H*_100%_ is a theoretical value that represents the estimated melt enthalpy of a fully crystalline PLA polymer, i.e., 93.7 J g^−1^ as reported in literature [69]. Finally, the term w_PLA_ represents the weight fraction of PLA on composites [70,71,72].

In addition to this thermal characterization, samples were analyzed by thermomechanical analysis (TMA) to obtain the dimensional stability of the obtained composites. In particular, the coefficient of linear thermal expansion (CLTE) was calculated from the slope of the characteristic TMA curves. To this, a thermomechanical analyzer TA Q400 (DE, USA) was used with a constant force of 20 mN. The heating program was set from –80 °C up to 100 °C at a heating rate of 2 °C min^−1^.

To better understand the thermomechanical behavior of the samples, they underwent dynamic mechanical thermal analysis (DMTA) in an oscillatory rheometer AR-G2 supplied by TA Instruments (New Castle, USA). A special clamp system designed for solid samples (40 × 10 × 4 mm^3^) were used, it works in torsion-shear conditions. The heating program was a temperature sweep from 30 °C up to 140 °C at a constant heating rate of 2 °C min^−1^. The maximum shear/torsion deformation (γ), and the oscillations frequency were defined as a percentage of 0.1% and 1 Hz, respectively.

### 2.4. Mechanical Properties

The most relevant information about mechanical properties was obtained by tensile, flexural, impact and hardness tests. Tensile and flexural tests were carried out as indicated in ISO 527 and ISO 178 respectively in a universal test machine ELIB 30 from S.A.E. Ibertest (Madrid, Spain). A load cell of 5 kN was used for both tests and the crosshead rate was set to 5 mm min^−1^. With regard to the impact properties, a Charpy pendulum with a total energy of 6 J from Metrotec S.A. (San Sebastián, Spain) was used on unnotched samples. The hardness was estimated using a Shore durometer with the “D” scale, model 673-D from J. Bot S.A. (Barcelona, Spain). All mechanical tests were run on five different samples and the average values of the corresponding properties were obtained.

### 2.5. Morphology Characterization

The morphology of the fractured samples (after failure in the impact test) was analyzed by field emission scanning electron microscopy (FESEM). A Zeiss Ultra55 FESEM microscope from Oxford Instruments (Oxfordshire, UK) working at an acceleration voltage of 2 kV was used. As PLA/HNTs composites are not electrically conducting, samples were subjected to a sputtering process with gold-palladium in a high vacuum sputter coater EM MED020 from Leica Microsystems (Milton Keynes, UK).

### 2.6. Water Uptake of PLA/HNTs

Water uptake of PLA/HNTs composites was carried out as indicated in ISO 62:2008 with distilled water at 30 ± 1 °C for a period of 98 days. Rectangular samples with dimensions 80 × 10 × 4 mm^3^ were initially dried at 60 °C for 24 h to remove residual moisture in an air circulating oven 2001245 DIGIHEAT-TFT from J.P. Selecta, S.A. (Barcelona, Spain). After this drying process, samples were submerged in distilled water and removed after planned times. These samples are dried with a cotton cloth and subsequently, are weighed in an analytic balance AG245 from Mettler-Toledo Inc. (Schwerzenbach, Switzerland); after weighting, samples were submerged aging in distilled water. The amount of absorbed water during the water uptake was calculated from Equation (2).
(2)$$\Delta m_{t}\left(\%\right)=\left(\frac{W_{t}-W_{0}}{W_{0}}\right)\times 100,$$
where *W*_*t*_ is the mass after a time *t* while *W*_0_ represents the dry weight of PLA/HNTs composites before the water uptake process. After a particular immersion time (saturation time) changes in water absorption can be neglected as they are extremely low. Then, it is possible to obtain the saturation mass that is denoted as Δmass.

### 2.7. Disintegration in Controlled Compost Soil

Biodegradation test, or more correctly, disintegration in the controlled compost soil test, was carried out following ISO 20200. The disintegration process was carried out at a temperature of 58 °C and a relative humidity on soil of 55%. Squared samples (20 × 20 mm^2^) and a thickness of 1 mm were buried into a biodegradation reactor, prepared as indicated in the corresponding standard. After some periods, samples were unburied, washed with distilled water and dried at 50 °C and, finally, weighed in an analytic thermobalance. The weight loss during the biodegradation process was calculated from Equation (3).
(3)$$Weight\;loss\;\left(\%\right)=\left(\frac{W_{0}-W_{t}}{W_{0}}\right)\times 100.$$

In this equation, *W_t_* represents the mass after a degradation time *t* and *W*_0_ stands for the dry weight of PLA/HNTs composites before disintegration.

## 3. Results and Discussion

### 3.1. Influence of HNTs Content on Mechanical Performance of PLA/HNTs Biocomposites

Table 2 shows a summary of the main mechanical parameters obtained from different tests, i.e., tensile, flexural, impact and hardness, as a function of increasing HNTs content. The first thing one can realize is that the tensile strength (σ_t_) was not highly affected by the presence of HNTs. It is true that we could observe a slight decrease in σ_t_ with increasing HNTs content. In particular, neat PLA showed a σ_t_ of 64.6 MPa while all composites with HNTs showed a σ_t_ comprised between 58–59 MPa, which represents a maximum percentage decrease of about 10%. On the other hand, the effect of HNTs on ductile properties such as elongation at break (%ε_b_) was much more pronounced. PLA is an intrinsically fragile polymer with a restricted elongation at break of only 6.1%. As we can see in Table 2, HNTs produced an embrittlement effect on PLA, leading to %ε_b_ lower than 4%. The maximum decrease corresponded to the PLA composite with 9 wt % HNTs with an elongation at break of 3.3% (which represents a percentage decrease of about 45%). HNTs were dispersed into the PLA polymer matrix and promoted a loss on material cohesion. Therefore, the stress transfer from the embedded particles to the surrounding matrix was restricted, and this had a negative effect on elongation at break. Similar findings have been reported in literature with PLA matrices [73,74]. As expected, the tensile modulus increased with increasing HNTs content. It is important to remember that the tensile modulus (*E*_t_) represents the ratio between the applied stress (σ) and the obtained elongation (ε) in the linear region of a stress-strain curve. As above-mentioned, both tensile strength and elongation at break decrease with increasing HNTs content, but the decrease in %ε_b_ was much more pronounced than that observed for σ_t_. As the strain was in the denominator, as the denominator (ε) decreased more than the numerator (σ), the overall effect was an increase in *E*_t_. While neat PLA was characterized by an E_t_ value of 2086 MPa and the composite with 9 wt% HNTs was more brittle, with a tensile modulus of 2311 MPa (>10% increase). These results are in accordance to those reported by Chen et al. [45]. This embrittlement could be related to an increase in the structural stiffness of PLA chains due to HNTs-PLA interactions [75,76]. These low decrease in tensile properties is related to the use of the PVAc compatibilizer, which is highly hydrophilic that can establish interactions with both hydroxyl groups (–OH) contained in PLA (end chains) and HNTs surface, thus leading to good embedding of the HNTs into the PLA matrix with a positive effect on mechanical performance. Pracella et al. [77] reported the interesting compatibilizing effect of poly(vinyl acetate), PVAc to enhance compatibilization in PLA/cellulose nanocrystals, as it provides increased adhesion between these two components.

With regard to flexural properties, the behavior of PLA-HNT composites followed the same tendency observed for tensile tests. All the herein developed composites with PLA and HNTs show increased flexural modulus (*E*_f_) compared to neat PLA. It is worthy to remark that PLA composites with 9 wt% HNTs offered a *E*_f_ value of 3927 MPa, which was noticeably higher than neat PLA (3570 MPa). As the above-mentioned, addition of HNTs provided increased structural stiffness on PLA polymer chains [75,76,78]. On the contrary, the flexural strength (σ_f_) decreased with increasing HNT content. At this point it is worthy to remark that deformation and strength were highly sensitive to material cohesion. Although some HNTs-PLA interactions are expected, presence of HNTs aggregates breaks cohesion and this has a negative effect on both properties in tensile or flexural conditions. Therefore, the flexural strength (sf) of neat was 116 MPa while this was reduced down to values of 107 MPa for PLA composites containing 9 wt % HNTs (around 7.7% decrease). Although it has been reported the reinforcing properties of HNTs on a PLA matrix, Therias et al. [68] have reported a slight decrease (almost negligible) of tensile strength with increasing HNTs content (up to 12 wt %), the most noticeable effect is an increase in stiffness. It is important to remark that the modulus represents the stress to strain ratio in the linear region of a tensile or flexural diagram. As the tensile strength remains with high values and elongation at break is remarkably reduced, then this ratio increases. Therefore, the tensile and flexural moduli increase with increasing HNTs content. In contrast, Chen et al. [79] have reported alternating values of tensile strength (increase and decrease) with different HNTs loading, but, in general, the stiffness is always increased with HNTs loading. It is worthy to remark the work by Guo et al. [80]. In their work they report the effect of pristine HNTs and alkali-modified HNTs. The show a decrease in tensile strength and elongation at break by increasing the loading of pristine HNTs but, in contrast, alkali-modified HNTs contribute to an increase in tensile strength, which is attributed to improved PLA/HNTs interactions due to the alkali treatment, which enhances more available hydroxyl groups in the outer surface.

Table 2 also summarizes Shore D hardness values for all composites, compared with neat PLA. All values were close to 82 and there was a very slight increase with the presence of HNTs. Nevertheless, this slight change was included in the deviation itself and, therefore, it was not possible to conclude a clear tendency.

Table 2 also includes the impact strength values obtained from the Charpy test. As expected, addition of HNTs provided an embrittlement effect as indicated by the modulus (in both tensile and flexural conditions). This embrittlement was also evident from the impact strength values. Neat PLA offered an impact strength of 1.46 kJ m^−2^ and this was reduced down to values close to half (0.71 kJ m^−2^) in composites with 9 wt% HNTs. Impact strength was also related to material cohesion and, as indicated previously, presence of HNTs aggregates led to poor material cohesion and this had a negative effect on both resistance (maximum stress that the material can withstand) and deformation, both parameters playing a key role in impact strength. The embedded HNTs particles acted as starting points for crack initiation and growth thus leading to a clear embrittlement, which can be observed by morphology analysis.

The morphologies of the fractured samples after impact tests are gathered in Figure 3 at different magnifications. Neat PLA (Figure 3a,b) showed a typical brittle fracture surface, with low roughness and a smooth fracture surface. As the HNTs loading increased, it was possible to detect an increase roughness, which was attributed to the presence of HNTs. HNTs promoted an embrittlement against impact, which favored crack formation and growth. This is because HNTs are highly hydrophilic and do not establish strong interactions with PLA matrix. At higher magnification (5000×) it was possible to see a fine particle dispersion of HNTs embedded into the PLA polymer matrix (Figure 3d,f,h). It is possible to observe the presence of HNTs aggregates as well as some individual HNTs embedded in the PLA matrix. As the HNTs content increased the density of these aggregates/individual HNTs increased. Although it seems these particles were fully embedded into the PLA matrix, there was poor interaction between them and the surrounding matrix even with the presence of PVA compatibilizer. This phenomenon provided poor material cohesion and, therefore, stress concentration phenomena could take place. Some research works indicate that HNTs usually give good particle dispersion due to the tubular structure if HNTs, which is responsible for a weakening of nanotube–nanotube interactions. In addition, weak interactions can be obtained between the PLA matrix and the embedded HNTs particles by hydrogen bonds between the carbonyl groups in PLA and the hydroxyl groups in halloysite [81,82].

Magnified FESEM images taken at 10,000× (Figure 4) showed in a clearer way, the presence of these aggregates and individual HNTs. 

As it can be seen in Figure 4a, good HNT dispersion is achieved for this relatively low HNT loading of 3 wt%. Individual HNTs can be detected as well as some aggregates (in de upper-right side). This aggregate formation is also detectable in PLA/HNTs composites containing 6 wt% HNTs (Figure 4b) located at the left side; nevertheless, the presence of individual well-dispersed HNTs can also be observed. Finally, Figure 4c shows the fracture surface corresponding to the PLA/HNT composite with 9 wt% HNTs. It is clearly detectable the presence of larger aggregates and some individual HNTs embedded in the PLA matrix. To check this dispersion Figure 5 gathers the FESEM image of a PLA/HNT composite (9 wt% HNT) and their corresponding energy-dispersive X-ray spectroscopy mapping images for oxygen carbon (C Kα1_2), aluminum (Al Kα1; O Kα1) and silicon (Si Kα1). As the above-mentioned, HNTs are aluminosilicate structures and, therefore, the presence of Al_2_O_3_ (Al Kα1) and SiO_2_ (Si Kα1 and O Kα1) was evidenced by the EDX analysis. In addition, these elements mapping suggest the presence of HNTs aggregates as the size was of 2.5 × 5 μm^2^ (see upper-left side of O Kα1, Al Kα1 and Si Kα1 EDX mapping images), as well as some individual points related to individual HNTs as observed by the FESEM analysis.

### 3.2. Influence of HNTs Content on Thermal Behaviour of PLA/HNTs Biocomposites

The effect of HNTs on thermal properties was assessed by differential scanning calorimetry (DSC). Figure 6 and Table 3 gathers the main parameters from DSC analysis for neat PLA and PLA-HNTs composites with increasing HNTs loading. With regard to the glass transition temperature (*T*_g_), the addition of HNTs did not provide any remarkable change in this parameter, which indicates poor HNTs-PLA interactions. In fact, the *T*_g_ values remained almost constant with values around 64 °C. Regarding the cold crystallization process, it is possible to observe a slight decrease in the cold crystallization peak temperatures (*T*_cc_) as observed by Prashanta et al. [74]. This is directly related to the nucleant effect that HNTs can exert in PLA chains, thus allowing PLA chains to fold in a packed way to form crystallites. The melt peak temperature (*T*_m_) was not affected by the presence of whatever loading of HNTs, remaining at values of about 172–173 °C. Regarding the degradation onset temperatures (*T*_d_), no clear effect of HNTs could be detected as all values were comprised between 322 and 327 °C [83]. With the obtained values of the melt and cold crystallization enthalpies (Δ*H*_m_ and Δ*H*_cc_) respectively, the percentage degree of crystallinity (%χ_c_) was calculated and it is shown in Table 3. As observed by Murariu et al. [84], addition of halloysite provides slightly lower crystallinity values. Table 3 shows two different crystallinity values, one is %χ_c_s_, which represents the degree of crystallinity of the injection molded sample without any removal of the thermal history (this takes into account the difference between the melt and cold crystallization enthalpies) and a second %χ_c_max_, which stands for the maximum crystallinity PLA can reach in these composites (this only considers the melt enthalpy). As it can be seen, this maximum crystallinity followed the same tendency, it means, a decrease with increasing HNT loading. This could be related to formation of less perfect crystals due to the presence of HNTs as reported by Chen et al. [79] with similar crystallinities for neat PLA of about 37% and lower values of 33–34% for a HNT loading of 10–15%. They also reported a clear decrease in the cold crystallization process as observed in this work, while very slight changes were observed for the glass transition temperature, T_g_, in accordance to the results in this research.

In addition to DSC characterization, the dimensional stability was assessed by thermomechanical analysis (TMA) through determining the coefficient of linear thermal expansion (CLTE) below and above the glass transition temperature (*T*_g_). Obviously, the CLTE was much lower at temperatures below the *T*_g_ compared to temperatures above *T*_g_. This is due to the glassy behavior of the material below the *T*_g_, as expected. On the contrary, above *T*_g_ the material becomes more plastic and this allows higher expansion. Nevertheless, the effect of HNTs on overall CLTE can be neglected as all the CLTE values for neat PLA and all its composites with HNTs changed in a very narrow range comprised between 76.1 and 78.7 μm m^−1^ °C^−1^. This identical tendency can be observed for the CLTE above the T_g_ with values comprised in the 130.5–133.7 μm m^−1^ °C^−1^ range. It has been reported the positive effect of HNTs on dimensional stability of [85] polyimide (PI) composites up 40–50 wt% loading of HNTs and polyamic acid (PAA). This high loading contributes to lowering the CLTE of polyimide (PI) films to be compatible with most metals, thus leading to polymer matrix composites with application as film capacitors. The obtained results in the present work suggest a slight improvement on the dimensional stability but it is not highly pronounced due to the relatively low HNTs loading compared to other works.

Dynamic-mechanical thermal analysis (DMTA) is very helpful to assess the effect of HNTs on both thermal and mechanical properties. Figure 7 shows the plot evolution of the storage modulus *(G′*; Figure 5a) and the dynamic damping factor (*tan* δ; Figure 7b). Regarding to neat PLA, its characteristic DMTA curve gave interesting information. Below 50 °C, *G′* remained almost constant and in the temperature, range comprised between 55–70 °C, a threefold decrease could be detected. This dramatic decrease in the storage modulus, *G′* was directly related to the temperature range in which the glass transition occurs. Another important process that could be observed by DMTA in neat PLA was the cold crystallization process. Crystallization is related to the formation of a packed-ordered structure, which involves an increase in the storage modulus, *G′*. Therefore, as we can see, *G′* increased in the temperature range of 80–100 °C. Regarding the effect of HNTs on DMTA behaviour, it is possible to see that all PLA-HNTs DMTA curves showed the same shape of that of neat PLA but with slight changes. On one hand, all *G′* curves were shifted to higher values thus indicating an increase in the storage modulus, *G′* which is directly related to the tensile and flexural modulus as described previously. Similar findings have been reported by Prashanta et al. [74]. Regarding the glass transition process, it seems that all curves were overlapped, which indicates slight or no changes in T_g_ as observed by DSC. On the other hand, regarding the cold crystallization process, DSC revealed a decrease in the peak temperature, and this is in total agreement with the DMTA curves of neat PLA and PLA-HNTs composites. It is possible to see in Figure 7a that the characteristic curves were moved to lower temperatures as the HNTs content increased. On the other hand, Figure 7b shows the evolution of the dynamic damping factor (*tan* δ) with the temperature. Despite that there are several methods to obtain the *T*_g_ from the DMTA analysis, one of the most accepted methods considers *T*_g_ as the peak maximum of the dynamic damping factor. The peak was located at 65 °C, which is in total agreement with the values obtained by DSC and there were no detectable changes in *T*_g_. It is possible to find a correlation with the maximum values of the dynamic damping factor with increasing HNTs loading since *tan* δ represents the ratio between the loss modulus (*G”*) and the storage modulus (*G′*). As indicated, *G′* increased with HNTs loading and as it is in the denominator of the dynamic damping factor, it led to a decrease in *tan* δ as it can be observed in Figure 4b. Table 4 shows a summary of some DMTA parameters taken at different temperatures, so that one can see in a clear way how the storage modulus of the samples behaved at different temperatures.

### 3.3. Study of the Water Uptake of PLA/HNTs Biocomposites

The evolution of the water uptake of PLA-HNTs composites is shown in Figure 8. As it can be seen, there was an increase in the absorbed water as a function of the immersion time and followed the typical Fick′s Law. An initial stage with high slope related to water absorption (∆mass) could be seen. In a second stage this increase was less pronounced and finally, a stationary mass was obtained (equilibrium water uptake), denoted as (∆mass_∞_).

The lowest water uptake values were observed for neat PLA with a saturation water content of 0.95% reached after 91 days. Halloysite nanotubes are highly hydrophilic materials due to the aluminosilicate structure and, consequently, as the HNTs loading increases, the saturation water increases as observed by Russo et al. [73]. In particular, the saturation for the PLA composite with 3 wt% HNTs was 1.31% while this was still higher for the composite containing 9 wt% HNTs (1.67%). The hydrophilic nature of HNTs was responsible for this increase. HNTs offer a high surface area with a high number of hydroxyl groups that contribute to the water uptake rate and equilibrium water. Therefore, as the HNTs loading increases, the ∆mass_∞_ also increases.

### 3.4. Disintegration in Controlled Compost Soil of PLA/HNTs Biocomposites

The weight loss of PLA-HNTs composites during the biodegradation (disintegration in controlled compost soil) is shown in Figure 9. All materials showed an incubation time of about nine days in which very slight weight changes took place. Above this induction time, all materials started a slow weight loss of up to 30 days and above this, the disintegration rate increased. Disintegration of PLA and other aliphatic polyesters was directly related to hydrolytic degradation of the ester groups; for this reason, it is necessary to maintain a relative humidity of 55% at a moderate temperature of 58 ± 2 °C to speed up the process. Neat PLA is the one with the highest disintegration rate as reported in literature [86,87]. After 35 days the weight loss was close to 30% and a 90% weight loss was achieved after 49 days. Above 49 days, PLA was fully disintegrated, and it was not possible to recover any piece to weight it. The presence of HNTs delayed the disintegration process. At 48 days, neat PLA was fully disintegrated while all PLA-HNTs composites showed a weight loss of about 80%. The overall effect of HNTs on the disintegration was a delay in the rate and lower weight loss after the same period compared to neat PLA as halloysite did not degrade due to its inorganic nature.

Figure 10 gathers some optical images of the disintegration process of neat PLA and PLA-HNTs composites with varying HNTs loading. Above two weeks, it was possible to appreciate a clear change in surface in almost all materials, which is representative of embrittlement (crack formation and growth), as Aguero et al. [88] reported, during the first two and three weeks, neat PLA is in the induction period. After four weeks all materials show some disintegration level. Neat PLA shows a less fragmented structure at this time. As it has been detected previously with the weight loss, presence of HNTs delay the disintegration process. After six weeks, the consistency of all materials has been lost and there is not a clear effect of the HNTs content, Paul et al. [89] have investigated the effect of organo-nanomodifiers in PLA blends in which it can be seen that they tend to degrade faster than neat PLA, but it is expectable that the final weight loss is directly related to the HNT content as this inorganic component does not degrade in these conditions, unlike this, in other researches [68], halloysite nanotubes have an effect that promotes degradation in photooxidation media.

## 4. Conclusions

This work assessed the technical viability of PLA composites with different halloysite nanotubes content (HNTs) for further uses. In particular, this work focused on the effect of HNTs loading (in the 3–9 wt% range) on mechanical, thermal, thermomechanical and disintegration properties of PLA-HNT composites. In general, the mechanical properties were slightly lower than those of neat PLA but the decrease in tensile or flexural strength was less than 7%, which was a positive effect. On the contrary, the elongation at break was reduced to a half with the presence of HNTs. Nevertheless, it is worthy to note that PLA itself was extremely brittle. With regard to thermal properties, presence of HNTs led to lowering the cold crystallization process but the glass transition temperature (*T*_g_) did not change. Identical behavior was obtained with differential scanning calorimetry (DSC) and dynamic-mechanical thermal analysis (DMTA) thus showing the consistency of the obtained results. Regarding the water uptake, the presence of highly hydrophilic HNTs contributes to increased water uptake (which is a negative effect on most industrial applications, but could be a positive effect on medical applications). Finally, the disintegration at PLA with HNTs was slightly delayed but, in general, all composites were almost disintegrated in a reasonable time. This work opens new possibilities to these composites for further applications in medicine as the overall properties are maintained and HNTs could be used as carriers for controlled drug delivery.

## Figures and Tables

**Figure 1 polymers-11-01314-f001:**
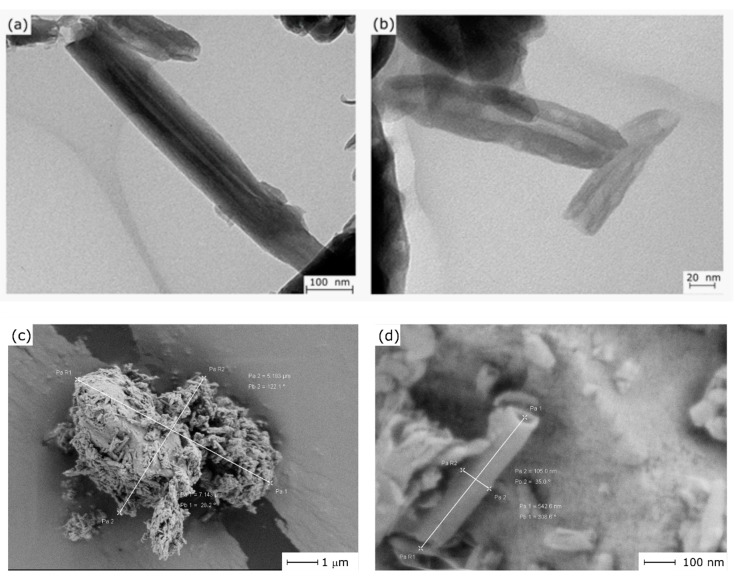
Different images showing the structure of halloysite nanotubes (HNTs). Transmission electron microscopy (TEM) images of halloysite nanotubes at different magnifications (**a**) 30,000×, (**b**) 80,000×. Field emission scanning electron microscopy (FESEM) images of (**c**) HNT aggregate at 10,000× and (**b**) isolated HNTs showing dimensions and the tubular structure at 100,000×.

**Figure 2 polymers-11-01314-f002:**
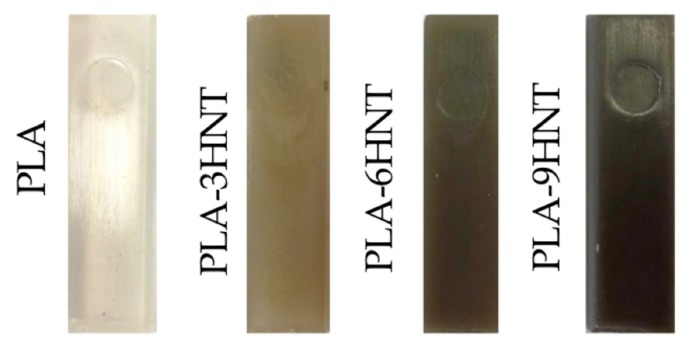
Digital images of PLA-HNT composites with different HNTs loading.

**Figure 3 polymers-11-01314-f003:**
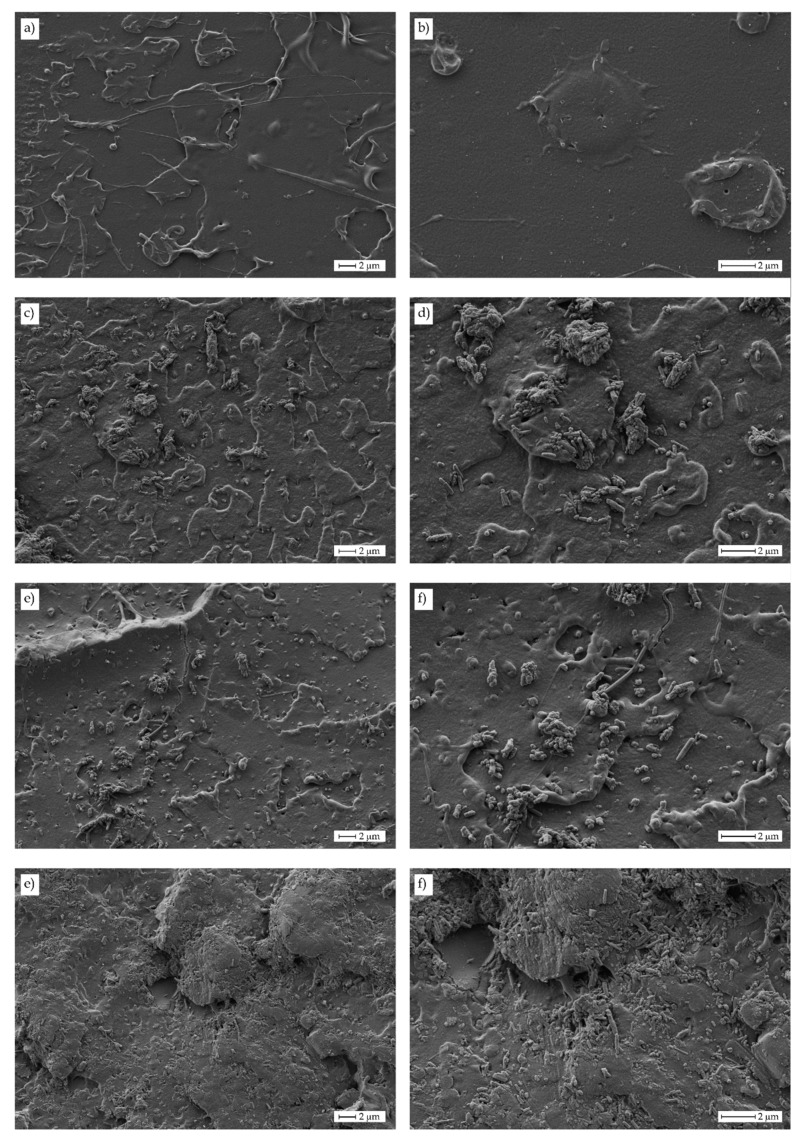
Field emission scanning electron microscopy (FESEM) images at different magnifications (2500×: left column; 5000×: right column) of fractured samples from impact tests corresponding to PLA-HNT composites with different HNTs loading. (**a**,**b**) neat PLA, (**c**) and (**d**) 3 wt% HNTs, (**e**) and (**f**) 6 wt% HNTs and (**g**) and (**h**) 9 wt% HNTs.

**Figure 4 polymers-11-01314-f004:**
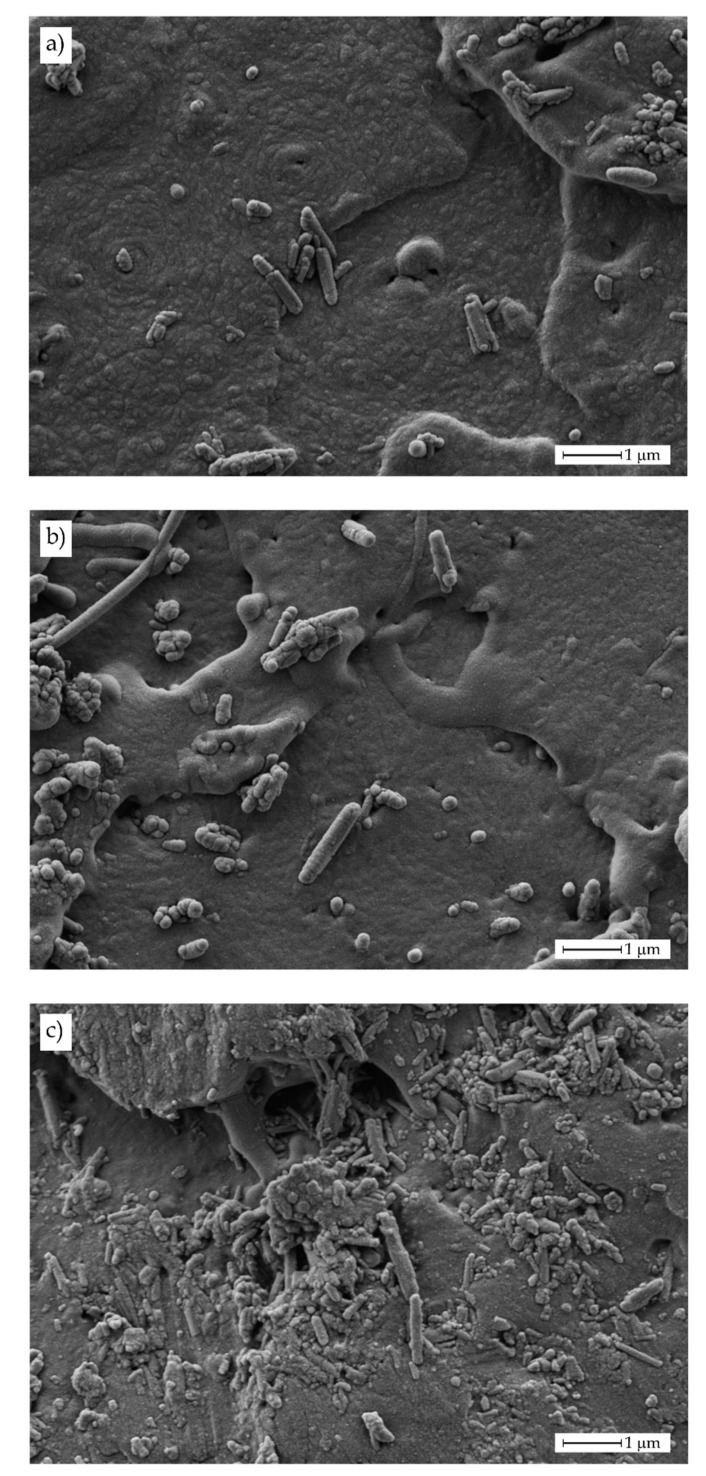
Field emission scanning electron microscopy (FESEM) images at 10,000× of fractured samples from impact tests corresponding to PLA-HNT composites with different HNTs loading. (**a**) 3 wt% HNTs, (**b**) 6 wt% HNTs and (**c**) 9 wt% HNTs.

**Figure 5 polymers-11-01314-f005:**
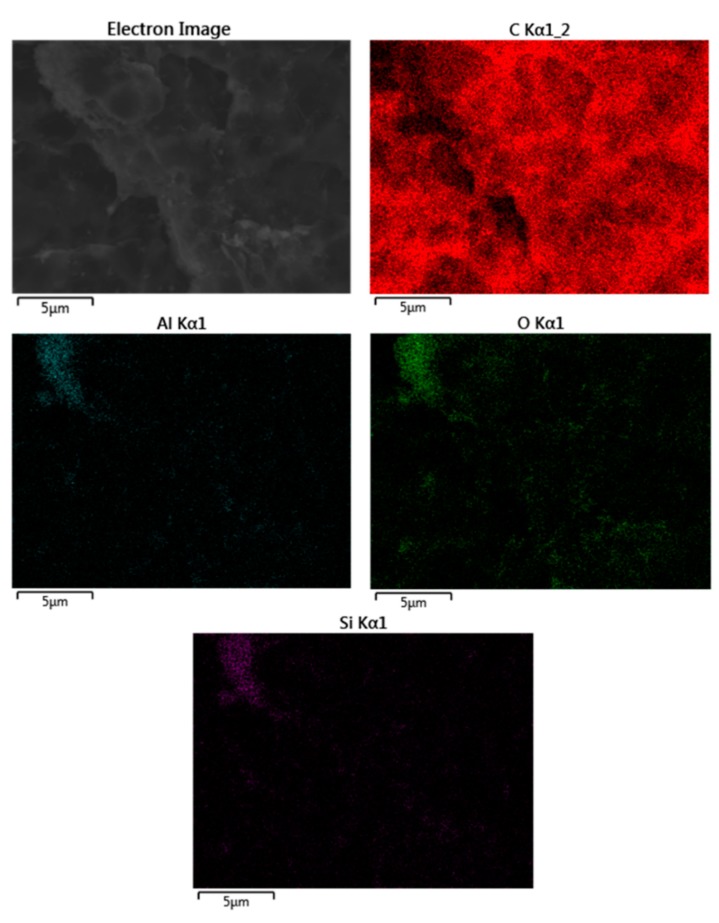
Field emission scanning electron microscopy (FESEM) image of a PLA/HNT composite with 9 wt% HNTs at 5000× and EDX mapping corresponding to carbon (C Kα1_2), aluminum (Al Kα1), oxygen (O Kα1) and silicon (Si Kα1).

**Figure 6 polymers-11-01314-f006:**
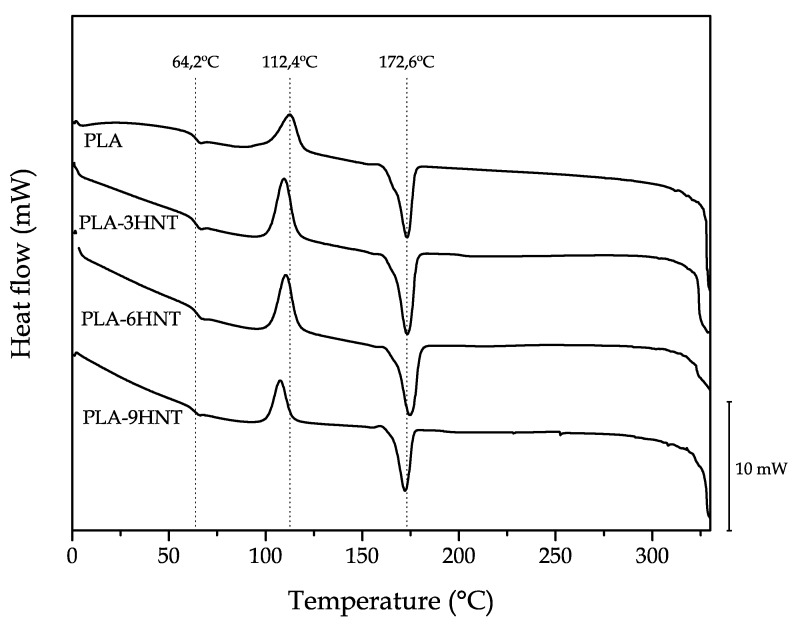
Comparative plot of the differential scanning calorimetry (DSC) thermograms corresponding to PLA-HNT composites with different HNTs loading.

**Figure 7 polymers-11-01314-f007:**
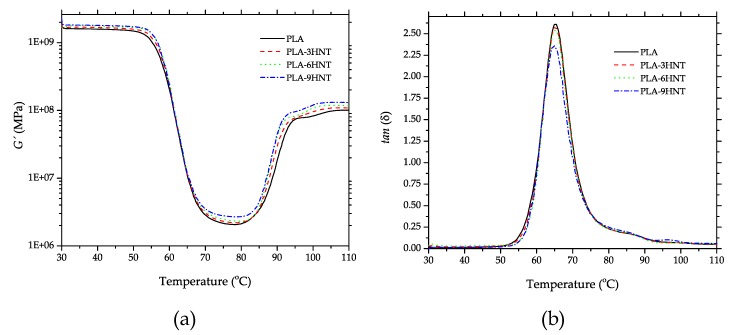
Plot evolution of the dynamic mechanical thermal properties (DMTA) for PLA-HNTs composites with different HNTs loading (**a**) storage modulus, *G′* and (**b**) dynamic damping factor, *tan* δ.

**Figure 8 polymers-11-01314-f008:**
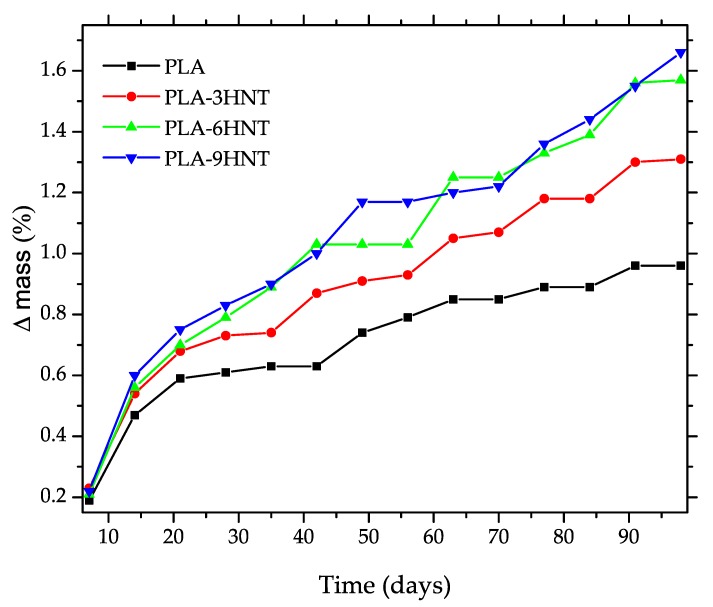
Evolution of the water uptake process in PLA-HNTs composites with different HNTs loading.

**Figure 9 polymers-11-01314-f009:**
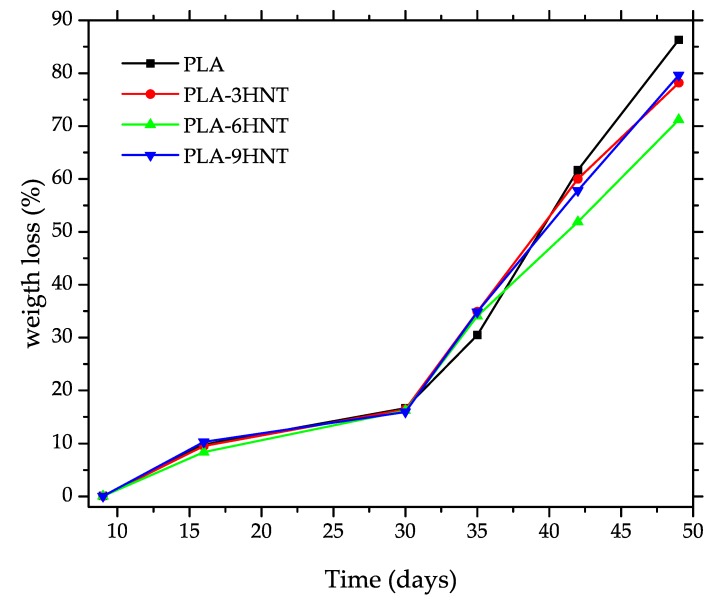
Follow up of the disintegration process in controlled compost soil of PLA-HNTs composites with different HNTs loading.

**Figure 10 polymers-11-01314-f010:**
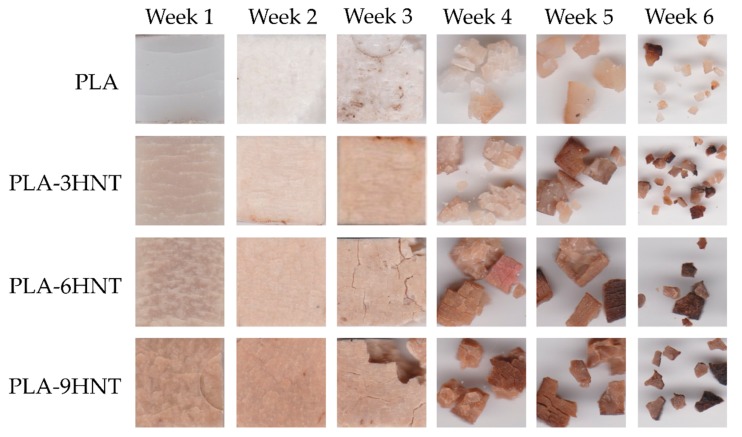
Optical images of the disintegration in controlled compost soil of PLA-HNT composites with different HNTs loading.

**Table 1 polymers-11-01314-t001:** Labeling and composition of poly(lactic acid) composites with different HNTs loadings with poly(vinyl acetate) compatibilizer.

Code	PLA (wt%)	HNTs (wt%)	PVAc (phr) *
PLA	100	-	-
PLA-3HNT	97	3	0.3
PLA-6HNT	94	6	0.6
PLA-9HNT	91	9	0.9

* phr represents the weight parts of additive per one hundred weight parts of the PLA/HNT composite material.

**Table 2 polymers-11-01314-t002:** Mechanical properties of PLA/HNTs composites obtained from tensile tests (tensile modulus—E_t_, tensile strength—σ_t_ and elongation at break—%ε_b_), flexural tests (flexural modulus—E_f_ and flexural strength—σ_f_), hardness (Shore D) and impact (impact strength from the Charpy test).

Code	Tensile			Flexural		Shore D Hardness	Impact Strength (kJ m^−2^)
	Modulus, E_t_ (MPa)	Strength, σ_t_ (MPa)	Elongation at Break (ε_b_%)	Modulus, E_f_ (MPa)	Strength, σ_f_ (MPa)		
**PLA**	**2086 ± 82**	64.6 ± 1.6	6.1 ± 0.8	3570 ± 39	116.1 ± 2.1	81.7 ± 1.0	1.46 ± 0.5
PLA-3HNT	2097 ± 73	59.2 ± 2.6	3.7 ± 0.5	3701 ± 100	105.5 ± 3.9	81.0 ± 2.0	1.18 ± 0.2
PLA-6HNT	2160 ± 78	58.7 ± 0.7	3.9 ± 0.1	3918 ± 252	110.4 ± 3.4	82.0 ± 2.0	1.03 ± 0.2
PLA-9HNT	2311 ± 132	57.7 ± 2.4	3.3 ± 0.2	3927 ± 128	107.6 ± 6.8	83.2 ± 2.7	0.71 ± 0.2

**Table 3 polymers-11-01314-t003:** Main thermal parameters of PLA-HNT composites with different HNTs loading obtained by differential scanning calorimetry (DSC) analysis.

Code	*T*_g_ (°C)	*T*_cc_ (°C)	ΔH_cc_ (J g^−1^)	T_m_ (°C)	ΔH_m_ (J g^−1^)	T_d_ (°C)	χ_c_s *_ (%)	χ_c_max **_ (%)
PLA	64.2 ± 3.2	112.4 ± 7.9	25.1 ± 2.6	172.6 ± 13.8	−33.7 ± 0.6	326.2 ± 6.9	9.2 ± 0.2	36.0 ± 0.8
PLA-3HNT	64.0 ± 2.7	109.6 ± 6.7	24.8 ± 3.1	172.6 ± 12.0	−32.3 ± 1.2	321.0 ± 18.5	8.2 ± 0.3	35.6 ± 1.1
PLA-6HNT	64.6 ± 4.5	106.2 ± 10.8	23.4 ± 1.2	174.0 ± 12.8	−29.2 ± 1.8	323.0 ± 23.7	6.6 ± 0.2	33.4 ± 0.9
PLA-9HNT	64.6 ± 4.0	105.4 ± 4.5	22.7 ± 2.6	171.6 ± 6.0	−28.6 ± 0.7	327.1 ± 20.4	6.9 ± 0.2	33.8 ± 1.0

* χ_c_s_ represents the crystallinity of the injection-molded materials without removing the thermal history. ** χ_c_max_ stands for the maximum crystallinity that can be obtained in PLA/HNT composites.

**Table 4 polymers-11-01314-t004:** Summary of some dynamic mechanical thermal properties of PLA-HNTs composites with different HNTs loading obtained by DMTA.

Code	G′ at 30 ℃	G′ at 50 ℃	G′ at 80 ℃	G′ at 120 ℃	Max *tan* δ
PLA	1599	1455	2.10	95.26	2.621
PLA-3 HNT	1678	1549	2.24	102.5	2.582
PLA-6 HNT	1731	1650	2.31	109.1	2.543
PLA-9 HNT	1843	1756	2.70	123.7	2.382

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
