# Peer review of "Manufacturing and Characterization of Functionalized Aliphatic Polyester from Poly(lactic acid) with Halloysite Nanotubes"

_polymers, 2019, doi:10.3390/polym11081314_

Round 1

Reviewer 1 Report

Montava-Jorda et.al. fabricated poly (lactic acid)/halloysite nanotube (HNTs) composite by injection molding method. The study regarding mechanical, thermal, and degradation of PLA/HNT composites are well presented. Before publication of this manuscript, the following concerns should be addressed.

1.     Abstract should be quantitative. Check it.

2.     The introduction part should be more informative with some PLA-based recent articles by 2019. Regarding PLA details, check following papers and cite it. Polymer Testing, 2017, 60, 132-139; Polymer degradation and stability, 2018 154, 248-260

3.     The digitals images of prepared composite poly (lactic acid)/halloysite nanotube (HNTs) should insert.

4.     The halloysite nanotube in composite should be verified by EDX. Please insert it.

5.     From Table 2, prepared composites showed lower the mechanical properties than neat PLA which limits application point of view. Justify it. The reason behind decrease in mechanical test should insert. Check it

6.     The author should insert DSC melting and cooling curve. TGA is also needed.

7.     Please provide higher magnification SEM images of HNT in the prepared composites.

8.     The inserted paragraph, from Line 364-371, should be a comparative study with published papers.

Author Response

1.     Abstract should be quantitative. Check it.

ANSWER

As indicated by Reviewer 1, quantitative information about the main results have been provided in the “Abstract” section to provide the overall effect of addition of HNTs to the PLA matrix.

2.     The introduction part should be more informative with some PLA-based recent articles by 2019. Regarding PLA details, check following papers and cite it. Polymer Testing, 2017, 60, 132-139; Polymer degradation and stability, 2018 154, 248-260

ANSWER

As recommended by the Reviewer 1, the references were updated to more recent published papers in 2018-19. In addition, we have read the recommended references in Polymer Testing and Polymer Degradation and Stability and added them in the “Introduction” section.

3.     The digitals images of prepared composite poly (lactic acid)/halloysite nanotube (HNTs) should insert.

ANSWER

As indicated by Reviewer 1, the digital images showing the appearance of PLA/HNTs composites has been added in the revised version. This is the new Figure 2 and, subsequently, all images have been re-numbered. In addition, we have compared the colour of the obtained materials with other similar materials reported in the literature.

4.     The halloysite nanotube in composite should be verified by EDX. Please insert it.

ANSWER

As indicated by the Reviewer, additional images have been provided in the revised manuscript. Moreover, we have used secondary literature about HNTs composition obtained by X-ray fluorescence spectroscopy by colleagues of our research group in a previous work. In the experimental section, two additional images have been added in Figure 1 to show FESEM images of a HNTs aggregate and an individual HNT with typical dimensions.

According to the Reviewer’s comments, additional FESEM images at higher magnifications have been obtained and provided in the revised version as the initially provided FESEM images at 100x and 500x were not able to detect HNTs. These new images at 2500x and 5000x give a clear evidence of the presence of both HNTs aggregates and individual HNTs to give support to the comments. Furthermore, presence of HNTs and their dispersion has been checked by EDX by showing the Al, Si, C, and O EDX mapping images.

5.     From Table 2, prepared composites showed lower the mechanical properties than neat PLA which limits application point of view. Justify it. The reason behind decrease in mechanical test should insert. Check it

ANSWER

We agree with the Reviewer’s comment. In fact, it is true that slightly lower tensile strength is obtained for PLA composites with increasing HNTs loading. This is the typical effect an inert filler produces in a polymer matrix. We have searched for secondary literature to give support to our results and we have found some controversy about the effects of HNTs on a PLA matrix. Most of the works report a slight decrease on tensile strength, an important decrease in elongation at break and a remarkable increase in stiffness, measured through the tensile or flexural moduli. These results are in total accordance to the obtained results in the present work and are mainly related to aggregate formation of HNTs due to their high hydrophilicity and poor polymer-particle interactions. Nevertheless, it is worthy to note that the main aim of this work is to assess the potential of these materials in terms of balanced properties for further research works focused on controlled additive release. In fact, in our research group, some colleagues have been working in a selective etching process to increase the lumen diameter of HNTs to act as load carriers. The decrease in tensile strength observed in this study is not remarkable (close to 10%) but the potential of the developed materials for 3D printing and controlled release of active compounds has been noticeably improved. In addition to this typical behaviour, some references with several treatments on HNTs have been commented and discussed in the text, showing the positive effect of a previous surface treatment or the use of a dispersant material to increase polymer-particle and avoid aggregate formation respectively. All these comments have been introduced in the “Results & discussion” section (mechanical characterization).

6.     The author should insert DSC melting and cooling curve. TGA is also needed.

ANSWER

As indicated by Reviewer 1, the DSC images have been inserted in the manuscript. These images show in a clear way the results gathered in table 3 regarding DSC main parameters. In particular, the invariable Tgand melt peak (Tm) and a shift of the cold crystallization to lower temperatures. Moreover, a new column in Table 3 has been added, showing the maximum degree of crystallinity that can be achieved in all PLA/HNT composites. To give support to the shown results, additional secondary literature with similar systems, has been provided and compared.

7.     Please provide higher magnification SEM images of HNT in the prepared composites.

ANSWER

We are in total accordance to this comment as the provided FESEM images do not allow identifying HNTs. As indicated by the reviewer, the low magnification FESEM images have been removed and new images at 2500x and 5000x have been provided. In addition, detailed FESEM images at 10000x show the presence of HNTs aggregates and individual HNTs. 

8.     The inserted paragraph, from Line 364-371, should be a comparative study with published papers.

ANSWER

As recommended by the Reviewer 1, this paragraph has included comparison of results found in literature with those obtained in the present research work.

Reviewer 2 Report

This work presents the concept of using HNT nanotubes for PLA modification. While this subject is up-to-date and interesting as presented in the introduction section, however, many changes are needed to cover the quality of the manuscript.

- The authors have used Ingeo 6201d, this grade of PLA is mainly used for the production of fibers, please comment/explain why this choice was made 

- Table 1 is presenting the composition of the prepared materials, the authors have used PVA as the compatibilizer, however the choice of this polymer has not been clearly explained. Did the authors prepare reference samples without the addition of PVA were made, this comparison would allow a clear answer regarding the meaningfulness of using a PVA compatibilizer,

- line 213...should be "..characterized...."

- in line 215, the authors indicate that the results of mechanical tests are consistent with the data obtained by Liu and Chen, however, in my opinion, especially the publication of liu points to other conclusions, such as increased strength and elongation at low HNT concentrations...please comment

- In the sample morphology section, the authors describe the visible nanotubes and their good dispersion. In my opinion, pictures that were prepared, can not testify to this, white circles indicated only the places of probable aggregates, but such a small magnification does not allow to confirm all the statements from the text ... to confirm the described phenomena, I recommend preparing TEM photos.

- In the DSC results section the authors discuss the shape of thermogram plots, however, no charts are included, only short table of the basic values is posted. In my opinion it would be necessary to include these graphs, especially due to the high temperature range, where the authors try to determine the degradation temperature

- the description of the results for the CLTE index analysis is very brief, please provide information if the measurement was conducted along or across the direction of the polymer flow. If the tests were performed "simply" on a more flat surface (in normal direction), then the CLTE changes will sometimes make the expansion factor worse, so the recommended orientation of the measurement is flow direction

- line 314 and 323, the authors describe the influence of temperature, but it seems to me that it is about the content of HNT

Author Response

- The authors have used Ingeo 6201d, this grade of PLA is mainly used for the production of fibers, please comment/explain why this choice was made 

ANSWER

It is true this PLA grade is intended for melt spinning of fibers due to its melt flow index. We have been working with a textile research institute in Alcoy (AITEX) in several projects related to fiber manufacturing and additivities. Nevertheless, due to the good melt flow index of this PLA grade, this material also gives good results processed by injection moulding as we have reported in other papers with new plasticizers and as base material for wood plastic composites with excellent performance. As recommended by the reviewer we have added a sentence indicating the suitability of this grade for injection moulding.

- Table 1 is presenting the composition of the prepared materials, the authors have used PVA as the compatibilizer, however the choice of this polymer has not been clearly explained. Did the authors prepare reference samples without the addition of PVA were made, this comparison would allow a clear answer regarding the meaningfulness of using a PVA compatibilizer,

ANSWER

As indicated by the Reviewer, additional comments on the use of poly(vinyl acetate) PVAc have been added. Moreover, new secondary literature to give support to this, has been provided in the revised version.

- line 213...should be "..characterized...."

ANSWER

As indicated by the Reviewer, this typo has been corrected. Additionally, we have carefully read the manuscript in-depth to detect and correct any other typos.

- in line 215, the authors indicate that the results of mechanical tests are consistent with the data obtained by Liu and Chen, however, in my opinion, especially the publication of liu points to other conclusions, such as increased strength and elongation at low HNT concentrations...please comment

ANSWER

We are in total agreement with the reviewer. Liu et al. reported a decrease in the elongation with higher HNT loads (>30) and a parallel increase in the tensile strength with increasing HNT loading. We have revised the comments in the text and corrected accordingly to their real findings.

- In the sample morphology section, the authors describe the visible nanotubes and their good dispersion. In my opinion, pictures that were prepared, can not testify to this, white circles indicated only the places of probable aggregates, but such a small magnification does not allow to confirm all the statements from the text ... to confirm the described phenomena, I recommend preparing TEM photos.

ANSWER

We are in total agreement with the Reviewer. These FESEM images give an idea of the micro-crack formation and growth but HNTs cannot be seen. For this reason, we have obtained additional FESEM images at higher magnifications and provided in the revised version since the initially provided FESEM images at 100x and 500x were not able to detect HNTs. These new images at 2500x and 5000x give a clear evidence of the presence of both HNTs aggregates and individual HNTs to give support to the comments. Furthermore, presence of HNTs and their dispersion has been checked by EDX by showing the Al, Si, C, and O EDX mapping images. In addition, a new FESEM image with a magnification of 10000x, shows in a clear way the presence of both individual HNTs and aggregates, thus giving support to the statements. The text has been corrected accordingly to these new provided FESEM and EDX images.,

- In the DSC results section the authors discuss the shape of thermogram plots, however, no charts are included, only short table of the basic values is posted. In my opinion it would be necessary to include these graphs, especially due to the high temperature range, where the authors try to determine the degradation temperature

ANSWER

As indicated by Reviewer 2, the DSC images were inserted in the manuscript and additional comments about the main thermal transitions observed by DSC have been provided.

- the description of the results for the CLTE index analysis is very brief, please provide information if the measurement was conducted along or across the direction of the polymer flow. If the tests were performed "simply" on a more flat surface (in normal direction), then the CLTE changes will sometimes make the expansion factor worse, so the recommended orientation of the measurement is flow direction

ANSWER

As indicated by Reviewer 2, the comments about CLTE have been extended by providing new comments and comparison of some results with other polymer/HNTs composites such as those of polyimide and HNTs, intended for film capacitors with up to 40-50 wt% loading.

- line 314 and 323, the authors describe the influence of temperature, but it seems to me that it is about the content of HNT

ANSWER

As suggested by Reviewer 2, it was a typo. It was double checked and rephrased to give a coherent paragraph

Round 2

Reviewer 1 Report

Author addressed all concerns regarding work. So, I recommend it for final publication.

Reviewer 2 Report

Most of my comments and suggestions have been included in the latest version of the article. In my opinion, the current version of the manuscript is suitable for publication in Polymers.